# GRIM19 Impedes Obesity by Regulating Inflammatory White Fat Browning and Promoting Th17/Treg Balance

**DOI:** 10.3390/cells10010162

**Published:** 2021-01-15

**Authors:** JooYeon Jhun, Jin Seok Woo, Seung Hoon Lee, Jeong-Hee Jeong, KyungAh Jung, Wonhee Hur, Seon-Yeong Lee, Jae Yoon Ryu, Young-Mee Moon, Yoon Ju Jung, Kyo Young Song, Kiyuk Chang, Seung Kew Yoon, Sung-Hwan Park, Mi-La Cho

**Affiliations:** 1The Rheumatism Research Center, Catholic Research Institute of Medical Science, The Catholic University of Korea, Seoul 137-040, Korea; jhunjy@catholic.ac.kr (J.J.); ulbojs@catholic.ac.kr (J.S.W.); jjeonghee1@gmail.com (J.-H.J.); Seonyeong@catholic.ac.kr (S.-Y.L.); shinhwadin7@catholic.ac.kr (J.Y.R.); moonym@catholic.ac.kr (Y.-M.M.); rapark@catholic.ac.kr (S.-H.P.); 2Division of Immunology, Department of Microbiology and Immunobiology, Harvard Medical School, Boston, MA 02115, USA; redcap817@catholic.ac.kr; 3Research Center, Impact Biotech, Seoul 137-040, Korea; yojikung@naver.com; 4The Catholic University Liver Research Center & WHO Collaborating Center of Viral Hepatitis, College of Medicine, The Catholic University of Korea, Seoul 137-040, Korea; wendyhur@catholic.ac.kr (W.H.); yoonsk@catholic.ac.kr (S.K.Y.); 5Division of Gastrointestinal Surgery, Department of General Surgery, Seoul St. Mary’s Hospital, The Catholic University of Korea, Seoul 137-040, Korea; leazl8012@gmail.com (Y.J.J.); skygs@catholic.ac.kr (K.Y.S.); 6Cardiovascular Center and Cardiology Division, Seoul St. Mary’s Hospital, College of Medicine, The Catholic University of Korea, Seoul 137-040, Korea; kiyuk@catholic.ac.kr; 7Department of Internal Medicine, Seoul St. Mary’s Hospital, College of Medicine, The Catholic University of Korea, Seoul 137-040, Korea; 8Division of Rheumatology, Department of Internal Medicine, Seoul St. Mary’s Hospital, College of Medicine, The Catholic University of Korea, Seoul 137-040, Korea

**Keywords:** obesity, GRIM19, STAT3, Th17

## Abstract

Obesity, a condition characterized by excessive accumulation of body fat, is a metabolic disorder related to an increased risk of chronic inflammation. Obesity is mediated by signal transducer and activator of transcription (STAT) 3, which is regulated by genes associated with retinoid-interferon-induced mortality (GRIM) 19, a protein ubiquitously expressed in various human tissues. In this study, we investigated the role of GRIM19 in diet-induced obese C57BL/6 mice via intravenous or intramuscular administration of a plasmid encoding GRIM19. Splenocytes from wild-type and GRIM19-overexpressing mice were compared using enzyme-linked immunoassay, real-time polymerase chain reaction, Western blotting, flow cytometry, and histological analyses. GRIM19 attenuated the progression of obesity by regulating STAT3 activity and enhancing brown adipose tissue (BAT) differentiation. GRIM19 regulated the differentiation of mouse-derived 3T3-L1 preadipocytes into adipocytes, while modulating gene expression in white adipose tissue (WAT) and BAT. GRIM19 overexpression reduced diet-induced obesity and enhanced glucose and lipid metabolism in the liver. Moreover, GRIM19 overexpression reduced WAT differentiation and induced BAT differentiation in obese mice. GRIM19-transgenic mice exhibited reduced mitochondrial superoxide levels and a reciprocal balance between Th17 and Treg cells. These results suggest that GRIM19 attenuates the progression of obesity by controlling adipocyte differentiation.

## 1. Introduction

Obesity is a complex metabolic disease related to multiple etiologies including nutritional, medicinal, and genetic factors [1]. Obesity is a public health problem that has resulted in the widespread occurrence of metabolic disorders such as cardiovascular disease, diabetes, and hypertension [2,3,4]. Multiple morbidities are involved in the development of obesity including significantly elevated blood cholesterol, glucose, and triglyceride levels in obese individuals compared with healthy controls [5,6]. Obesity prevention and therapies are essential for reducing the incidences of these morbidities. Obesity mediates mitochondrial dysfunction and disrupts various metabolic profiles. Genes can be associated with retinoid-interferon-induced mortality (GRIM) 19, a 16-kDa protein, which is primarily recognized as a nuclear protein associated with apoptosis [7]. GRIM19 activity is associated with mitochondrial respiration. A deficiency in GRIM19 has been associated with exacerbation of mitochondrial respiratory chain activity [8,9].

Excessive adipogenesis is associated with obesity progression, and the inhibition of white adipocyte differentiation is crucial for preventing this progression. Inhibition of the differentiation of 3T3-L1 cells reduces the accumulation of fat tissue and attenuates metabolic parameters in vitro [10,11]. In 3T3-L1 cells, signal transducer and activator of transcription (STAT) 3 plays a key role in adipogenesis by inducing adipocyte differentiation [12,13]. Obesity is characterized by an enhanced inflammatory response and the differentiation of pathogenic T cells, such as T helper (Th) 17 cells [14]. The loss of STAT3 in T cells reduces Th17 and regulatory T (Treg) cell populations in mouse models of obesity [15]. In addition, STAT3 deficiency in T cells may provide a therapeutic effect in terms of ameliorating excessive inflammation, glucose tolerance, and insulin sensitivity [15]. Notably, STAT5 deficiency promotes adiposity and maintains lipid homeostasis [16]. Moreover, the absence of STAT5 in white adipose tissue (WAT) improved glucose metabolism parameters [16]. Thus, reciprocal regulation of STAT3 and STAT5 is important for WAT metabolism.

However, GRIM19 exerts inhibitory effects on STAT3 activity and may suppress its activation [17]. GRIM19 reduced activation at the mRNA level in various STAT3-responsive genes; moreover, the loss of GRIM19 promoted the expression of STAT3-responsive genes [18]. Recent publications have also shown that GRIM19 induces a reciprocal balance in Th17/Treg cells [19].

Here, we hypothesized that GRIM19 attenuates obesity through the inhibition of adipogenesis. This study investigated whether GRIM19 can ameliorate the progression of obesity. We investigated the role of GRIM19 in adipogenesis both in vivo and in vitro. We also examined the therapeutic activity of GRIM19 in vivo in a high-fat diet (HFD)-induced mouse model of obesity. Finally, we examined the effects of GRIM19 on the Th17/Treg balance, which is modulated by the STAT3 pathway, in our mouse model of obesity.

## 2. Materials and Methods

### 2.1. Ethics Statement

The Animal Care Committee of The Catholic University of Korea approved the experimental protocol (permit number: CUMC-2014-0017-01). All animal handling procedures and protocols followed the guidelines of the Animal Research Ethics Committee of the Catholic University of Korea and the US National Institutes of Health Guidelines.

### 2.2. Animals

Four-week-old C57BL/6 mice (Orient Bio, Seongnam, Korea; *n* = 10 mice) were maintained under specific pathogen-free conditions and fed a diet consisting of standard laboratory mouse chow (Ralston Purina, St. Louis, MO, USA) that contained 60 Kcal fat-derived calories and water *ad libitum*. All surgeries were performed under isoflurane anesthesia, and all available efforts were made to minimize suffering.

### 2.3. Transgenic (Tg) DNA Sample Preparation and Generation of Tg Mice

To generate GRIM19-Tg mice, a GRIM19 fragment (based on the open reading frame in mice) was synthesized by GenScript Corporation (Piscataway, NJ, USA) with codon optimization for expression in mammalian cells. The fragment was inserted into a pcDNA3.1+ vector (Invitrogen, Carlsbad, CA, USA) containing a cytomegalovirus promoter. The insertion point was at the *Hind*III and *Xho*I restriction sites located upstream of the cytomegalovirus promoter. After sufficient quantities of DNA had been obtained in the bacterial expression line, the ~3.8 kb backbone fragment was removed by gel extraction, and the ~2.0 kb fragment was used for microinjection. Tg mice overexpressing GRIM19 were generated against a C57BL/6 background and maintained by Macrogen Inc. (Seoul, Korea) via microinjection of 4 ng/µL transgene DNA directly into the male pronucleus of the zygote using a micromanipulator. Microinjected embryos were incubated at 37 °C for 1–2 h. Next, 14–16 injected one-cell-stage embryos were surgically transplanted into the oviducts of pseudo-pregnant recipient mice. After the F0 generation was born, tail-cutting samples were genotyped for the presence of the transgene and confirmed using polymerase chain reaction (PCR) analysis. The genomic DNA-confirmed GRIM19-Tg founder mice were then mated with C57BL/6 mice.

### 2.4. GRIM19 Vector Treatment

C57BL/6 mice received hydrodynamic injections of 100 µg of the GRIM19-overexpression vector or a mock vector. The vectors were also injected intramuscularly with 100 µg into the thigh area by electroporation at 2-week intervals [20]. The intramuscular injection was performed with a 31-G insulin syringe.

### 2.5. RNA Interference 

To knock down the expression of GRIM19, small interfering RNA (siRNA) specific for GRIM19 (sc-60766, Santa Cruz Biotechnology, Dallas, TX, USA) or scrambled siRNA (sc-37007, Santa Cruz Biotechnology) were nucleoporated using X-tremeGENE HP Transfection reagent (Roche, Mannheim, Germany). The siRNA (50 nM) (sequence) or scrambled siRNA was transfected in accordance with the manufacturer’s recommendations. After transfection, 3T3L-1 cells were incubated at 37 °C for 3 days and stimulated with differentiation medium.

### 2.6. Immunoblotting

Cells were lysed with RIPA Lysis and Extraction Buffer (89901, Thermo Fisher, Waltham, MA, USA) including Halt^TM^ Protease Inhibitor Cocktail (78410, Thermo Fisher) and protein concentrations were determined using the Bradford method (Molecular Devices, Downingtown, PA, USA). Protein samples were separated using 10% sodium dodecyl sulfate–polyacrylamide gel electrophoresis and transferred to Hybond membranes (Amersham Pharmacia Biotech, Piscataway, NJ, USA). Proteins were incubated with antibodies against GRIM19 (ab110240, Abcam, Cambridge, MA, USA), phospho-STAT3 (Tyr705; 9145, Cell Signaling, Danvers, MA, USA), phospho-STAT3 (Ser727; 9134, Cell Signaling), STAT3 (Cell 9139, Signaling), and β-actin (sc-47778, Santa Cruz Biotechnology). They were then detected using an enhanced chemiluminescence detection kit (Pierce, Rockford, IL, USA) and HyperFilm (Agfa, Mortsel, Belgium), with β-actin used as a loading control.

### 2.7. Gene Expression Analysis Using Real-Time PCR

PCR amplification and analysis were performed on a LightCycler 2.0 system (Roche Diagnostics, Mannheim, Germany) using version 4.0 software. All reactions were performed using LightCycler Fast Start DNA master SYBR green I (Takara, Shiga, Japan), in accordance with the manufacturer’s instructions. The following primers were used: *CCAAT/enhancer binding protein alpha* (forward: CAAGAACAGCAACGAGTACCG, reverse: GTCACTGGTCAACTCCAGCAC); *angiotensinogen* (forward: AACACCAGCATCCAGTTCAA, reverse: GGTTCAGTAGGCCATTCCTC); *adiponectin* (forward: GTCAGTGGATCTGACGACACCAA, reverse: ATGCCTGCCATCCAACCTG); *resistin* (forward: AAGAACCTTTCATTTCCCCTCCT, reverse: GTCCAGCAATTTAAGCCAATGTT); *adipocyte protein 2* (forward: GATGCCTTTGTGGGAACCT, reverse: CTGTCGTCTGCGGTGATTT); *pantothenate kinase 3* (forward: TGCTGTAGTGTCCCATTTCTGCCT, reverse: AGCTGGAACAGCAACACCTAGGAA); *PRDM16* (forward: GACATTCCAATCCCACC, reverse: CACCTCTGTATCCGTCAGCA); *fibroblast growth factor 21* (forward: CCTCTAGGTTTCTTTGCCAACAG, reverse: AAGCTGCAGGCCTCAGGAT); *cytochrome C oxidase polypeptide 7A1 (Cox7a1)* (forward: AGAAAACCGTGTGGCAGAGA, reverse: CAGCGTCATGGTCAGTCTGT); *lipoprotein lipase* (forward: GGAAGAGATTTCTCAGACATCG, reverse: CTACAATGACATTGGAGTCAGG); *leptin* (forward: CCTCATCAAGACCATTGTCACC, reverse: TCTCCAGGTCATTGGCTATCTG); *cell death-inducing DNA fragmentation factor alpha-like effector A* (forward: GCCGTGTTAAGGAATCTGCTG, reverse: TGCTCTTCTGTATCGCCCAGT); *fatty acid elongase 3* (forward: CGGGTTAAAAATGGACCTGA, reverse: CCAACAACGATGAGCAACAG); *peroxisome proliferator-activated receptor gamma coactivator 1-alpha (PGC1a)* (forward: GTCAACAGCAAAAGCCACAA, reverse: TCTGGGGTCAGAGGAAGAGA); *uncoupling protein 1 (UCP1)* (forward: CTTTGCCTCACTCAGGATTGG, reverse: ACTGCCACACCTCCAGTCATT); *cytochrome C1* (forward: GCTACCCATGGTCTCATCGT, reverse: CATCATCATTAGGGCCATCC); and *p53* (forward: CACGTACTCTCCTCCCCTCA, reverse: CTCCGTCATGTGCTGTGACT).

### 2.8. Flow Cytometry

Cytokine expression in mice was assessed via intracellular staining with the following antibodies: fluorescein isothiocyanate-conjugated anti-interleukin-17 (Ebio17b7; eBioscience, San Diego, CA, USA), phycoerythrin-conjugated anti-Foxp3 (FJK-16s; Thermo Fisher Scientific, Waltham, MA, USA), PerCP-Cyanine 5.5-conjugated anti-CD4 (RM4-5; Thermo Fisher Scientific), and allophycocyanin-conjugated anti-CD25 (PC61; BioLegend, San Diego, CA, USA). Cells were stimulated for 4 h with phorbol myristate and ionomycin, followed by the addition of GolgiStop (BD Biosciences, San Diego, CA, USA). The cultured cells were surface-labeled for 30 min and permeabilized with Cytofix/Cytoperm solution (BD Biosciences). Cells were stained intracellularly with fluorescent antibodies before analysis with fluorescence-activated cell sorting flow cytometry (i.e., FACSCalibur). For the measurement of mitochondrial reactive oxygen species, cells were stained with MitoSOX Red for 15 min at 37 °C and washed with phosphate-buffered saline, then analyzed using fluorescence-activated cell sorting. Data were collected and analyzed with FlowJo software (Tree Star, Ashland, OR, USA). Cells subjected to live cell imaging were maintained under culture conditions and stained with MitoSOX. Live cell imaging was performed using a fluorescence-based live cell imaging system (LSM 510 Meta; Carl Zeiss, Oberkochen, Germany) with the appropriate filters and lasers.

### 2.9. Confocal Microscopy

For immunostaining, 7 µm tissue sections of spleen were stained using phycoerythrin-conjugated anti-CD4 (GK1.5; BioLegend), fluorescein isothiocyanate-conjugated anti-CD4 (RM4-5; BD Biosciences), phycoerythrin-conjugated anti-interleukin-17 (Ebio17B7; eBioscience), allophycocyanin-conjugated anti-CD25 (PC61; BioLegend), phycoerythrin-conjugated anti-Foxp3 (FJK-16s; Thermo Fisher Scientific), phycoerythrin-conjugated anti-pSTAT3 (Tyr705, 4/P-STAT3; BD Biosciences), phycoerythrin-conjugated anti-pSTAT3 (Tyr727, 49/P-STAT3; BD Biosciences), and allophycocyanin-conjugated anti-pSTAT5 (SRBCZX; Thermo Fisher Scientific). The stained sections were examined using a Zeiss microscope (LSM 510 Meta; Carl Zeiss) at 400× magnification.

### 2.10. Glucose and Insulin Tolerance Tests

Mice were fasted overnight and subjected to intraperitoneal injection with glucose (1 g/kg body weight). Blood glucose levels were measured with an Accu-Chek Performa instrument (Roche, Basel, Switzerland) using whole blood taken from cut tail tips immediately before and at 30, 60, 90, and 120 min after the injection of glucose (Sigma-Aldrich, St. Louis, MO, USA). The insulin tolerance test was performed on nonfasted mice by intraperitoneally injecting the mice with insulin (1 U/kg body weight; Eli Lilly, Indianapolis, IN, USA). Blood glucose levels were measured before and at 30, 60, 90, and 120 min after insulin injection, using the Accu-Chek Performa instrument.

### 2.11. Preparation of Serum Samples for Biochemical Analyses

Blood samples were collected in serum tubes from treated and control mice at 14 weeks and stored at −70 °C until use. The levels of total serum cholesterol were measured using commercial kits (Wako Co., Osaka, Japan). Aspartate aminotransferase, alanine aminotransferase, glucose, free fatty acid, triglyceride, high-density lipoprotein-cholesterol, and low-density lipoprotein-cholesterol levels were measured using commercial kits from Asan Pharmaceutical Co. (Hwaseong, Korea). The serum levels were measured using a Hitachi 7600 analyzer (Hitachi, Tokyo, Japan). To investigate the effect of fasting on blood glucose levels in mice, blood samples were obtained from the tails. Glucose levels were then measured using the Accu-Chek instrument (Roche). Blood glucose was measured randomly from the experimental mice.

### 2.12. Diet and Temperature Studies

The HFD was administered *ad libitum* from weaning until indicated. Cold exposure involved animals in a single housing fed an HFD at 4 °C for 1 day, after which individual fat tissues were isolated.

### 2.13. Cell Culture and Induction of Differentiation

The 3T3-L1 and C2C12 cells were purchased from the Korean cell line bank and maintained in Dulbecco’s modified Eagle medium containing 10% fetal calf serum in an incubator with 5% CO_2_ at 37 °C. Murine 3T3-L1 preadipocytes were cultured in preadipocyte medium (PM-1-L1, ZenBio, Durham, NC, USA, 24-well plates, 1.2 × 10^4^ cells/well) and allowed to reach 100% confluence. The cells were then incubated for an additional 48 h before initiation of differentiation. Subsequently, the cells were incubated in an appropriate volume of 3T3-L1 cell differentiation medium (DM2-L1, ZenBio) for 3 days, followed by incubation with 3T3-L1 adipocyte maintenance medium (AM-1-L1, ZenBio). For C2C12 cells, T3 and insulin were added to the medium at concentrations of 1 and 20 nM, respectively, until the cells reached 75–80% confluence. Next, 3-isobutyl-1-methylxanthine (500 μM), dexamethasone (2 mM), and indomethacin (125 μM) were added to the medium overnight to induce differentiation. The cells were incubated in growth medium with T3 and insulin, with medium changes at 48 h intervals.

### 2.14. DNA Transfection

GRIM19-overexpression vector or a mock vector were transfected into 3T3-L1 and C2C12 cells using X-treme GENE HP Transfection reagent (Roche), in accordance with the manufacturer’s instructions.

### 2.15. Immunohistochemical Staining of Adipose and Liver Tissues

Adipose and liver tissues were fixed in 10% formalin for 24 h, embedded in paraffin, and cut into 5 µm sections. Deparaffinized and dehydrated sections were stained with Oil Red O and hematoxylin and eosin.

### 2.16. Measurement of Intracellular Triglyceride Content

Liver tissue was homogenized in 1 M NaCl and extracted twice in a 2:1 (*v*/*v*) chloroform/methanol solution. Hepatic lipids were extracted with 5% Triton X-100. After supernatants had been centrifuged, they were transferred to clean tubes and stored at −80 °C until use. Triglyceride levels were measured using an EnzyChrom™ triglyceride assay kit (Bioassay Systems, Hayward, CA, USA) in accordance with the manufacturer’s instructions. The data are expressed as µg lipid/mg cellular protein [21].

### 2.17. Statistical Analysis

All data are expressed as the means ± standard deviations (SDs). Experimental values are presented as the means ± SDs of triplicate cultures and are representative of experiments performed on three occasions. Statistical significance was determined via the Mann–Whitney U test or analysis of variance followed by a Bonferroni post hoc test using GraphPad Prism (version 5.01; GraphPad Software, La Jolla, CA, USA). Values of *p* < 0.05 were considered to indicate statistical significance.

## 3. Results

### 3.1. Overexpression of GRIM19 Reduces Expression of White Fat Genes and Induces Browning of WAT in 3T3-L1 Cells

As the conversion of WAT into brown adipose tissue (BAT) is a key factor in obesity therapy [22], we investigated the role of GRIM19 in the differentiation of adipocytes and expression of WAT- and BAT-related genes. In this study, 3T3-L1 cells were transfected with a plasmid encoding GRIM19. Overexpression of GRIM19 led to enhanced differentiation of 3T3-L1 cells in vitro, as shown in Figure 1A. Furthermore, GRIM19 downregulated mRNA levels of WAT-related genes including *CCAAT/enhancer binding protein alpha*, *angiotensinogen*, *adiponectin*, *resistin*, *adipocyte protein 2*, and *pantothenate kinase 3*, as shown in Figure 1B, whereas it promoted mRNA expression levels of BAT-related genes such as *PRDM16*, *fibroblast growth factor 21*, and *Cox7a1*, as shown in Figure 1C. Furthermore, transcript levels of *lipoprotein lipase* and *leptin* were reduced in GRIM19-overexpressing cells and elevated in GRIM19-depleted cells, as shown in Figure 1D. These data suggest that GRIM19 is a key factor for the browning of WAT.

### 3.2. GRIM19 Regulates BAT Differentiation

To investigate the effects of GRIM19 on BAT differentiation, C2C12 cells were transfected with a plasmid encoding GRIM19 or siRNA targeting GRIM19. Overexpression of GRIM19 led to enhanced differentiation of C2C12 cells, whereas depletion of GRIM19 led to reduced differentiation of C2C12 cells in vitro, as shown in Figure 2A. Transcript levels of BAT-related genes were increased in C2C12 cells overexpressing GRIM19, whereas transcript levels of BAT-related genes were downregulated in GRIM19-depleted C2C12 cells, as shown in Figure 2B,C. In addition, overexpression of GRIM19 led to elevated transcript levels of *p53* and *PRDM16*, as shown in Figure 2D. These data suggest that GRIM19 regulates brown adipogenic differentiation through *p53* and *PRDM16*.

### 3.3. GRIM19 Overexpression Inhibited the Progression of HFD-Induced Obesity

To determine the role of GRIM19 in the development of obesity, we fed an HFD to mock-vector and GRIM19-overexpressing mice to induce obesity. The control plasmid and plasmid encoding GRIM19 were injected into HFD-fed C57BL/6 mice. The HFD-fed mock-vector (mock HFD) mice exhibited a gradual increase in body weight. Conversely, HFD-fed mice overexpressing GRIM19 (GRIM19 HFD) gained less body weight than did mock HFD mice, as shown in Figure 3A. In addition, glucose levels in the serum were significantly reduced in GRIM19 HFD mice during glucose tolerance and insulin tolerance tests, as shown in Figure 3B, whereas serum aspartate aminotransferase and alanine aminotransferase levels were downregulated in GRIM19 HFD mice, compared with mock HFD mice, as shown in Figure 3C. We found that GRIM19 expression levels were elevated in the splenocytes of GRIM19-Tg mice compared with wild-type (WT) mice, as shown in Figure 3D. The HFD-fed GRIM19-Tg (GRIM19-Tg HFD) mice exhibited a phenotype similar to that of GRIM19 HFD mice. The GRIM19-Tg HFD mice gained less body weight, as shown in Figure 3E, and had lower levels of metabolic indicators of obesity (e.g., serum glucose, aspartate aminotransferase, and alanine aminotransferase) compared to HFD-fed WT (WT HFD) mice, as shown in Figure 3F.

### 3.4. GRIM19 Overexpression Improves Liver Function in Obese Mice

Abnormal liver function is related to the progression of obesity because obesity exacerbates the development of fatty liver [23]. GRIM19 ameliorates intracellular lipid accumulation by regulating the gene expression involved in fatty acid biosynthesis [24]. We conducted liver function experiments using serum from WT HFD and GRIM19-Tg HFD mice. The infiltration of immune cells into the liver and induction of fatty liver were attenuated in GRIM19-Tg HFD mice, compared with WT HFD mice. In addition, the production of hepatic triglycerides was reduced in the liver of GRIM19-Tg HFD mice, compared with the liver of WT HFD mice, as shown in Figure 4A. Electron microscopy revealed that lipid droplets were smaller in the liver of GRIM19-Tg HFD mice, as shown in Figure 4B. Histological analyses revealed smaller adipocytes in the epididymal fat of GRIM19-Tg HFD mice compared with WT mice, as shown in Figure 4C. The weight of epididymal fat was reduced in GRIM19-Tg HFD mice, compared with WT HFD mice, as shown in Figure 4D. Furthermore, lipid vesicles in the brown fat of GRIM19-Tg HFD mice were smaller. Compared with WT HFD mice, GRIM19-Tg HFD mice exhibited elevated expression levels of BAT-related genes, such as *UCP1*, *Cristae*, *PGC1a*, and *Cox7a1*, as shown in Figure 4E.

### 3.5. GRIM19 Regulation of the Reciprocal Balance Between Th17 and Treg in Obese Mice

Interleukin-17 and pSTAT3 expression levels were suppressed in spleen tissue from GRIM19-Tg mice, compared with WT mice. However, Treg cell numbers and pSTAT5 expression levels in spleen tissue were significantly elevated in GRIM19-Tg mice, compared with WT mice, as shown in Figure 5A. The expression levels of pSTAT3 (Tyr705) and pSTAT3 (Ser727) in GRIM19-Tg mice were slightly reduced, as shown in Figure 5B.

### 3.6. GRIM19 Overexpression Caused Thermogenic Effects in Obesity

We examined the proportions of interscapular brown and white fat pads in obese mice. Compared with WT HFD mice, GRIM19-Tg HFD mice had more interscapular brown fat pads, but fewer interscapular white fat pads, as shown in Figure 6A. We also measured the expression levels of BAT-related genes in the epididymal fat of obese mice exposed to 4 °C for 16 h. The transcript levels of BAT-related genes were significantly elevated in the epididymal fat of GRIM19-Tg HFD mice, compared with WT HFD mice, as shown in Figure 6B.

## 4. Discussion

The activity of GRIM19 has been investigated extensively in the context of apoptosis [7], and recent studies have reported its association with inflammatory autoimmune diseases [19]. However, few studies have investigated the role of GRIM19 in obesity. Here, we examined the therapeutic function of GRIM19 in obesity.

BAT/WAT differentiation effects seen in vitro are associated with GRIM19 effect in vivo on body weight. Dysregulated accumulation of WAT is presumably involved in the development of obesity, activating an inflammatory process in the obese state [25,26]. By contrast, various reports have highlighted BAT as a potential therapeutic target for pharmacological agents designed to treat obesity, thus improving the metabolic profile [27,28,29]. Previous reports have suggested that inhibiting the differentiation of 3T3-L1 cells contributes to significant obesity attenuation [10,11]. The conversion of WAT into BAT has a clear role in the treatment of obesity [22]. Our results demonstrated that GRIM19 controls the differentiation of C2C12 cells and inhibits the differentiation of 3T3-L1 cells, thus modulating reciprocal regulation of the WAT/BAT balance. These findings suggest that GRIM19-focused treatment may be a novel therapeutic strategy for treating obesity.

Obesity leads to chronic and severe inflammation. Adipocytes in individuals with obesity cause metabolic syndrome and inflammatory responses by regulating the release of adipocytokines [30]. Moreover, Th17 cell differentiation and inflammatory processes are upregulated in the obese state [14]. STAT3 transcriptional activity, a key factor in pathogenic Th17 cells, controls the inflammatory response, whereas STAT5 controls differentiation into Treg cells [31,32]. GRIM19 reduces STAT3 activation and ameliorates experimental autoimmune disease mediated by STAT3 [17,19]. In this study, GRIM19 reduced Th17 differentiation and pSTAT3 expression, whereas it enhanced Treg differentiation, thus upregulating pSTAT5 in HFD-fed obese mice. These results suggest that GRIM19 can impede the progression of obesity by regulating the reciprocal Th17/Treg balance through the suppression of STAT3 expression and induction of STAT5 expression.

The induction of BAT can be used to treat obesity. The browning of white fat exerts a therapeutic effect against the development of obesity [22]. Brown and beige adipocytes have generated interest in health science research due to their capacities to counteract metabolic diseases such as obesity and type 2 diabetes. Whereas some metabolic disorders and obesity are commonly linked to type 2 diabetes, BAT suppresses type 2 diabetes features. In addition, the level of p53, which can be reduced by GRIM19 deficiency, is essential for the differentiation of BAT in diet-induced obesity [33,34]. *PRDM16*, regulated by p53, plays an important role in the development of BAT [35,36]. It has been suggested that *fatty acid elongase 3* is essential for BAT onset, and gene expression levels of *PGC1α*, *Cox7a1*, *UCP1*, and *fibroblast growth factor 21* are also elevated in BAT [37]. In the present investigation, GRIM19 promoted BAT differentiation and gene expression of *p53* and *PRDM16*. The mRNA levels of *fatty acid elongase 3*, *PGC1α*, *Cox7a1*, *UCP1*, and *fibroblast growth factor 21* were upregulated in GRIM19-Tg mice. Thus, GRIM19 may be an important molecule for the treatment of obesity.

Because the metabolic variations involved in obesity elevate the risk of fatty liver and fat accumulation in the body [38,39], fatty liver and metabolic profiles are key factors in the development of obesity. Obesity promotes the elevation of triglyceride, aspartate aminotransferase, and alanine aminotransferase levels, concomitant with a reduction in high-density lipoprotein production [40,41,42]. The inhibition of low-density lipoprotein, triglycerides, and blood glucose also improves obesity-mediated metabolic disorders [43,44]. Our study showed that various metabolic profiles, including levels of triglycerides, aspartate aminotransferase, blood glucose, low-density lipoprotein-cholesterol, and fat accumulation in the liver were reduced, whereas *Cristae* was upregulated in GRIM19-Tg mice, compared with WT mice. These results suggest that GRIM19 can impede the progression of obesity by attenuating obesity-induced fatty liver and metabolic dysfunction. GRIM19 inhibits the progression of obesity by regulating BAT differentiation and the Th17/Treg balance. Because GRIM19 reduces STAT3 activation and inflammation, the conversion of WAT into BAT may be involved in the promotion of p53 and attenuation of mitochondrial dysfunction. In vitro and in vivo studies are needed to further validate the browning of white fat mediated by STAT3 inhibition. Our observations suggest that the therapeutic effects of GRIM19 ameliorate obesity by means of BAT induction and Th17 cell suppression through pSTAT3 inhibition. The results of this study indicate that GRIM19 may be a promising therapeutic candidate for the treatment of obesity.

## Figures and Tables

**Figure 1 cells-10-00162-f001:**
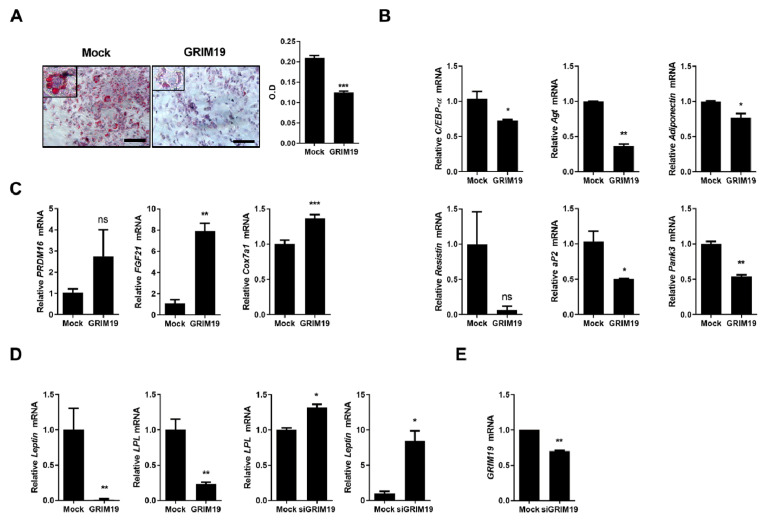
Genes associated with retinoid-interferon-induced mortality 19 (GRIM19) regulates the transcript levels of genes associated with white adipose tissue (WAT) and brown adipose tissue (BAT). Scale bar, 100 µm. (**A**) Representative images of Oil Red O-stained 3T3-L1 cells. Mock and GRIM19-overexpressing 3T3-L1 cells were stained with Oil Red O. (**B**,**C**) Transcript levels of genes associated with WAT (**B**) and BAT (**C**) in 3T3-L1 cells expressing control plasmids (MOCK) and plasmids encoding GRIM19 (Grim19). (**D**) Transcript levels of *leptin* and *lipoprotein lipase* (*LPL*) in 3T3-L1 cells expressing control plasmids (MOCK), plasmids encoding GRIM19 (Grim19), or GRIM19-depleted siRNA (siGrim19). (**E**) *GRIM19* mRNA levels were determined using real-time polymerase chain reaction. GAPDH was used as the internal control. Data are presented as the means ± standard deviations (SDs) of three independent experiments. * *p* < 0.05, ** *p* < 0.01, *** *p* < 0.001, and ns = not significant.

**Figure 2 cells-10-00162-f002:**
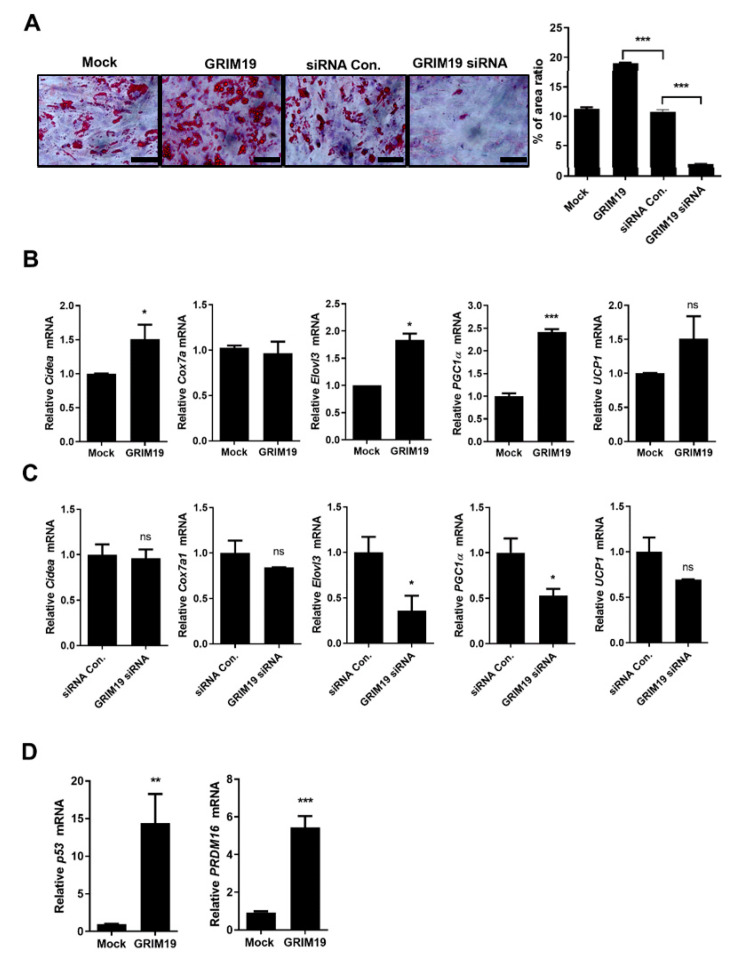
GRIM19 regulates BAT differentiation. (**A**) Representative images of Oil Red O-stained C2C12 cells. Mock, GRIM19-overexpressing, control siRNA (siRNA Con), or GRIM19 siRNA C2C12 cells were stained with Oil Red O. Scale bar, 100 µm. The bar graph shows averaged percentage of Oil Red O-stained area. (**B**) Transcript levels of *cell death-inducing DNA fragmentation factor alpha-like effector A* (*Cidea*), *cytochrome C oxidase polypeptide 7A1 (Cox7a1)*, *fatty acid elongase 3* (*Elovl3*), *PGC1a*, and *UCP1* in mock-vector (MOCK) and GRIM19-overexpressing C2C12 cells. (**C**) Transcript levels of *Cidea*, *Cox7a1*, *Elovl3*, *PGC1a*, and *UCP1* in control and GRIM19-depleted C2C12 cells. (**D**) Transcript levels of p53 and *PRDM16* in mock-vector and GRIM19-overexpressing C2C12 cells. GAPDH was used as the internal control. Data are presented as the means ± SDs of three independent experiments. * *p* < 0.05, ** *p* < 0.01, *** *p* < 0.001, ns = not significant.

**Figure 3 cells-10-00162-f003:**
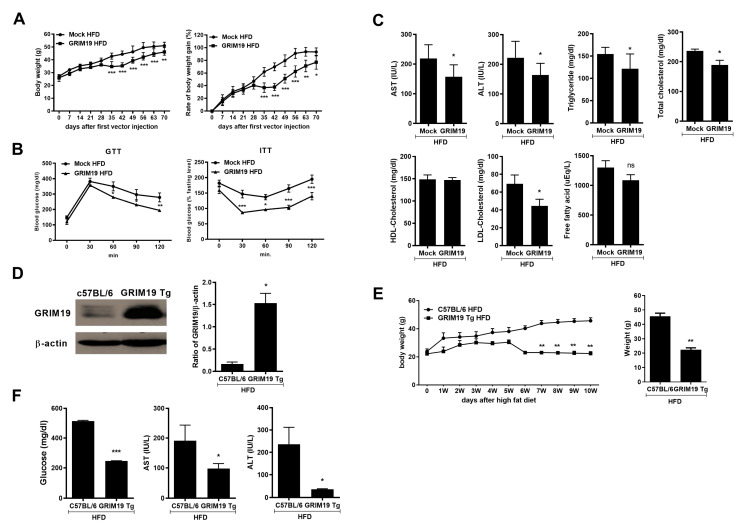
Overexpression of GRIM19 by two different strategies leads to the attenuation of high-fat diet (HFD)-induced obesity. (**A**) Kinetics of body weight changes (left) and rates of body weight gain (right) in HFD-fed control and GRIM19-administered mice over three independent experiments. (**B**) Glucose tolerance test (GTT, left) and insulin tolerance test (ITT, right) results for HFD-fed control and GRIM19-administered mice (*n* = 5). (**C**) Measurements of serum aspartate aminotransferase (AST), alanine aminotransferase (ALT), triglycerides, total cholesterol, high-density lipoprotein (HDL)-cholesterol, low-density lipoprotein (LDL)-cholesterol, and free fatty acid levels in HFD-fed control and GRIM19-administered mice (*n* = 5). (**D**) Splenocytes were obtained from wild-type (WT) and GRIM19-Tg mice. Cells were prepared to examine the expression of GRIM19. The bar graph shows relative intensity. (**E**) Kinetics of body weight changes in HFD-fed control and GRIM19-Tg mice. The bar graph shows the averaged body weights of HFD-fed WT and GRIM19-Tg mice at 10 weeks (*n* = 5–7). (**F**) Measurements of glucose, AST, and ALT levels in serum in HFD-fed control and GRIM19-Tg mice (*n* = 5–7). Data are presented as the means ± SDs. * *p* < 0.05, ** *p* < 0.01, *** *p* < 0.001, ns = not significant.

**Figure 4 cells-10-00162-f004:**
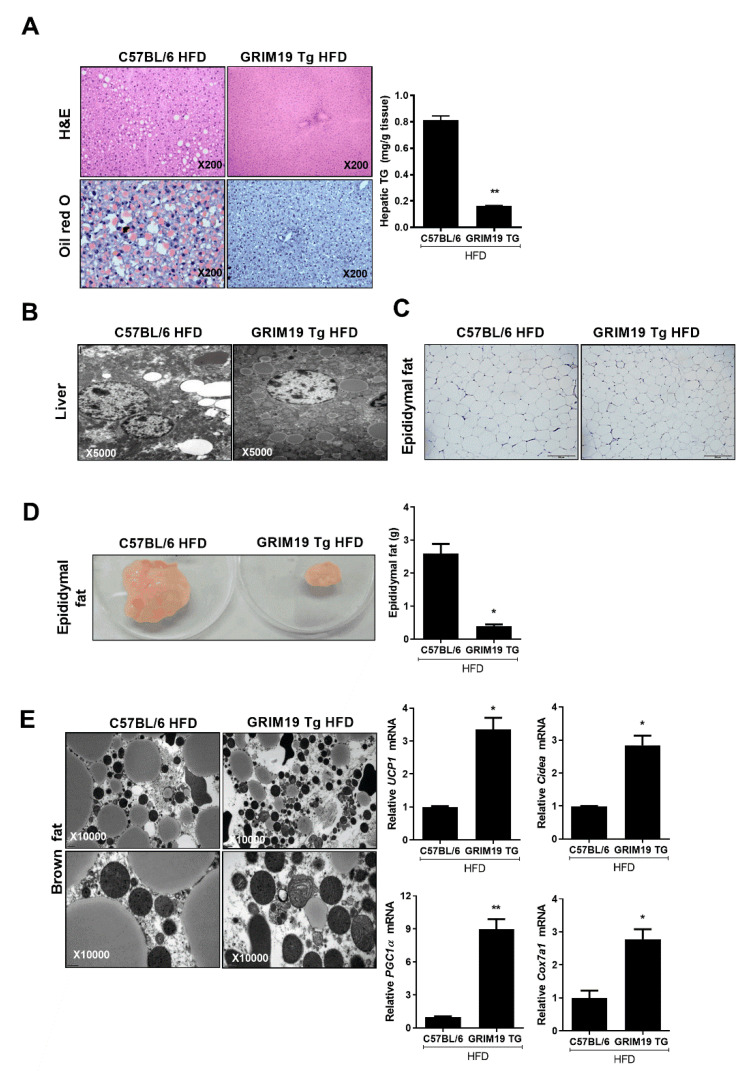
GRIM19 protects against diet-induced obesity and hepatic steatosis as well as regulates the fat profile. (**A**) Representative hematoxylin and eosin (H&E)- and Oil Red O-stained images of liver tissues from HFD-fed WT and GRIM19-Tg mice. The bar graph shows the averaged hepatic levels of triglycerides. Scale bar, 200 µm. (**B**) Representative electron microscopy images of liver tissue from HFD-fed WT and GRIM19 mice. Scale bar, 2 µm. (**C**) Representative H&E-stained images of the epididymal fat pad from HFD-fed WT and GRIM19-Tg mice (×200). (**D**) Representative images of epididymal fat from normal-fat diet (NFD)- and HFD-fed WT mice and HFD-fed GRIM19-Tg mice. The bar graph shows the averaged epididymal fat weight of HFD-fed WT and GRIM19-Tg mice. (**E**) Representative electron microscopy images of interscapular brown fat from HFD-fed WT and GRIM19-Tg mice. Scale bar, 2 µm. Bar graphs show the transcript levels of *UCP1*, *Cidea*, *PGC1a*, and *Cox7a1*. The transcript levels were normalized to that of GAPDH. Data are presented as the means ± SDs. * *p* < 0.05 and ** *p* < 0.01.

**Figure 5 cells-10-00162-f005:**
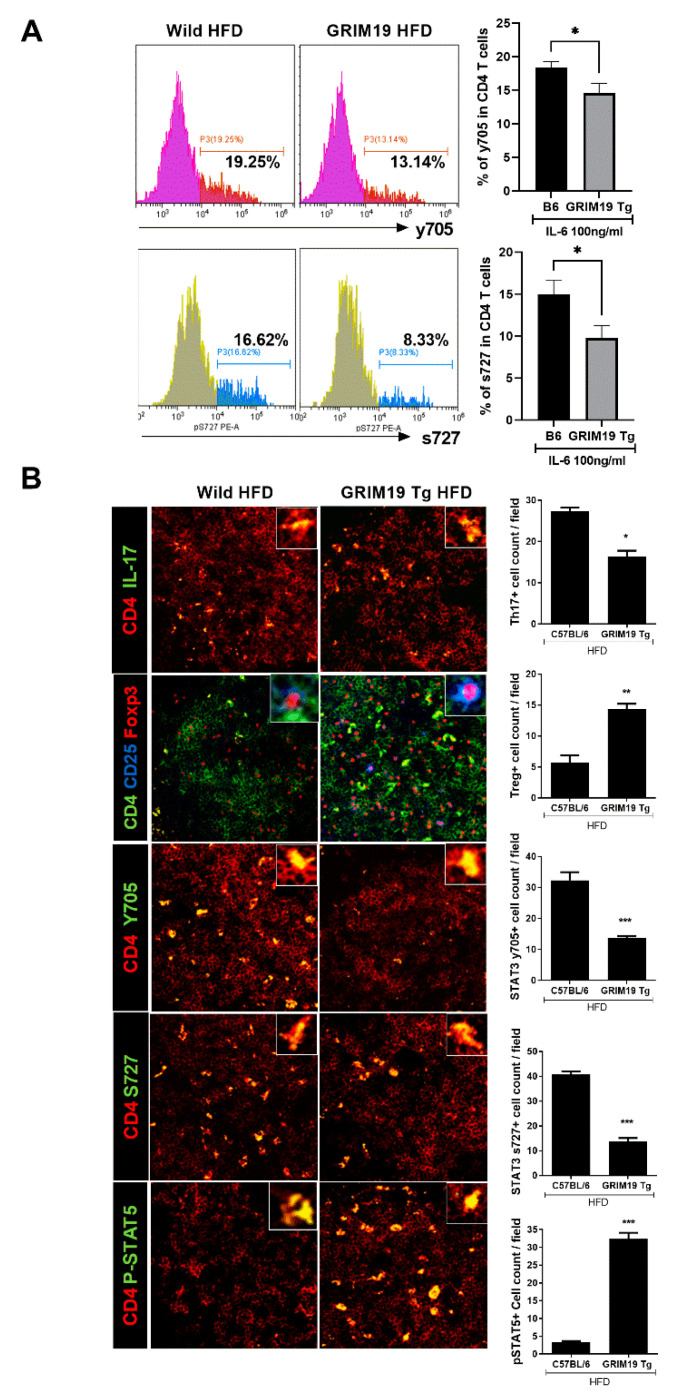
Effect of GRIM19 on Th17/Treg cell regulation in obese mice. (**A**) Representative flow cytometry plots showing the frequencies of CD4^+^pSTAT3 (Tyr705)^+^ and CD4^+^pSTAT3 (Ser727)^+^ cells in spleens from HFD-fed WT and GRIM19-Tg mice. Bar graphs show frequency of each cell type. (**B**) Representative confocal microscopy images of CD4^+^IL-17^+^ cells and CD4^+^CD25^+^Foxp3^+^, as well as CD4^+^pSTAT3 (Tyr705)^+^, CD4^+^pSTAT3 (Ser727)^+^, and CD4^+^pSTAT5^+^ cells in spleens from HFD-fed WT and GRIM19-Tg mice. Bar graphs show the numbers of each cell type in four independent quadrants. Data are presented as the means ± SDs of three independent experiments. Scale bar, 20 µm * *p* < 0.05, ** *p* < 0.01, *** *p* < 0.001.

**Figure 6 cells-10-00162-f006:**
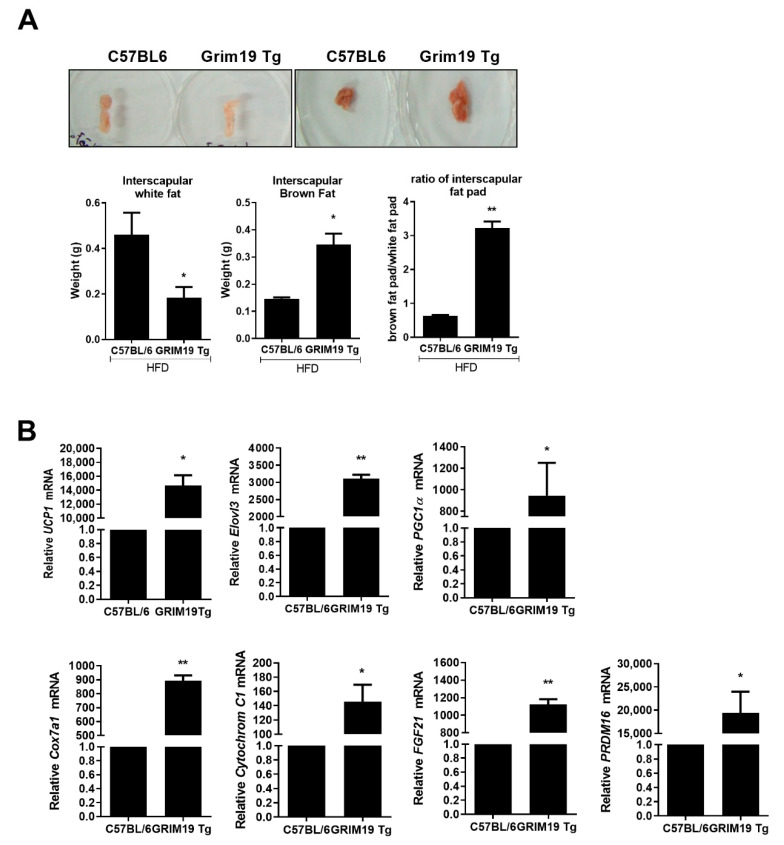
GRIM19-mediated anti-adipogenic and thermogenic effects in GRIM19-Tg and WT mice. (**A**) Representative images of interscapular white fat and interscapular brown fat from HFD-fed WT and GRIM19-Tg mice (top). Bar graphs show the averaged weights of interscapular white fat and interscapular brown fat, and ratios of interscapular white fat weight to that of interscapular brown fat in interscapular fat pads (bottom). (**B**) Transcript levels of *UCP1*, *Elovl3*, *PGC1a*, *Cox7a1*, *cytochrome C1*, *fibroblast growth factor (FGF) 21*, and *PRDM16* in epididymal fat pads from HFD-fed WT and GRIM19-Tg mice. The transcript levels were normalized to that of GAPDH. Mice were housed at 4 °C for 1 day before the experiments. Data are presented as the means ± SDs of three independent experiments. * *p* < 0.05 and ** *p* < 0.01.

## Data Availability

The data presented in this study are available in this article.

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
