# Peer review of "GRIM19 Impedes Obesity by Regulating Inflammatory White Fat Browning and Promoting Th17/Treg Balance"

_cells, 2021, doi:10.3390/cells10010162_

Round 1
Reviewer 1 Report
In this study, the authros show that Grim19 overexpression improves whole body glucose and insulin metabolism in high fat diet challenged mice along with a reduced body weight gain, which are novel functions of Grim19. In addition, liver function was preserved. Intriguingly, gonadaly adipose tissue mass was reduced while brown adipose tissue mas was increased suggesting that Grim19 may affect HFD-induced weight gain by regulating white adipocyte browning and/or BAT activity. However, several concerns exist that must be addressed exerimentally
Major comments:
Comment 1:
„Overexpression of GRIM19 Reduces Expression of White Fat Genes and Induces Browning of WAT in 3T3-L1 Cells“
The experimental proof fort he above mentioned sentence is extremely weak based on the data shown in Fig1 for several reasons: 1) 3T3-l1 cells cannot be converted into BAT cells i.e. these cells eare a poor model to test brwoning of white adipocytes. 2) The statement in lines 240-243 is oversimplified because the genes Cebpa, AdipoQ are ubiquitously expressed in WAT AND BAT and not specific for WAT, as ststated in the sentence: „The results showed that GRIM19 downregulated mRNA levels of WAT related genes including CCAAT/enhancer binding protein alpha, angiotensinogen, adiponectin, resistin, adipocyte protein 2, and pantothenate kinase 3 (Figure 1A), whereas it promoted mRNA expression levels of BAT-related genes such as PRDM16, fibroblast growth factor 21, and Cox7a1 (Figure 1B)“ Moreover, the classical BAT gene and best indicator for increased thermogenes is Ucp1 – what about effetcs of Grm10 modulaion (overexpression/silencing) on Ucp1?
Actually, the authors already have the best model to determine induction of brownining of WAT, which is their inguinal adipose tissues from Grim19-tg mice either on HFD or acutely subjected to 4°C.
Comment 2:
The authors show reduced Grim19 mRNA levels following transfection of 3T3-L1 cells with a siRNA targeting Grm19, however, the authors do not show successful Grm19 overexpressing in cells transfected with a Grim19 plasmid. The authors must show Grim19 mRNA overexpression folloing transfection with a Grim19 plasmid on the RNA as well a son the protein level by western blotting versus cells transfected with an empty plasmid. In addition Grim19 protein levels need tob e determined following transfection of 3T3-L1 cels with their Grim19 targeting siRNA versusu control siRNA
The authors show that Grim19 positively reguates C2C12 differentiation by assessing Oil-redO staining – instead of showing a rather small microscopic view oft he differentiated cells, it would be more convincing to show a fotograph of the whole well.
Most importantly, when assessing potential effects of Grim19 on (white/brown) adipocyte differentiation, PPARg, the master regulator of adipogenesis must be measured
Comment 3:
The authors claim that Grim19 injection or trangenic Grim19 expression counteracts obesity, however, the bodyweight shown in Figure 3E, and to a lesser extent Figure 3A in cntrol animals seems to decline suggesting some sort of animal discomfort/stress in the control group – and this seems to happen right when the body weights between experimental and control groups start to separate – this must be explained (Why do Grim19-tg animals show a decrease in body weight gain between days 28-35 followed by a plateu-phase in bw gain between days 35 and 42)?
What does the body weight gain curve look lime of Grim19-tg versus mock-plasmid treated animals on control diet for days 0-70? I.e can Grim19 overexpression also affect body weight independent of diet?
Comment 4:
The GRIM19 Vector Treatment needs tob e explaine din much more detail: i.e. after injection into the „thigh arear“ – how was the electroporation done? What instrument was used for this and how was this done exactly? Give a reference.
Comment 5:
The authors show that Grim19 regulates glucose metabolism in vivo – this is a very important finding – but what ist he molecular mechanism? The authors show that Grim19 (modestly) regulates Stat3 activity, at least in spleen, however – more pertinent to their glucose & insulin handling data shown in figure 3b would be an assessment of proxiaml insulin signaling in the liver and/or muscle and maybe fat (e.g. Akt phosphorylation as an indicator of insulin signaling).
In line with the above comment and the data shwoing improved liver function, less liver fat content in Grim19-tg mice: are these effects cell-autonomous? I.e. Is Grim19 in hepatocytes or muscle cells or adipocytes affecting directly lipid/glucose metabolism?
Comment 6:
The authors must show grim19 protein levels in liver and muscleat leats in Grim19-tg mice versus control mice, not just spleenocytes
Comment 7:
Lines 390-393 „We observed that 390 the expression levels of cyclooxygenase IV and mitotracker were enhanced, whereas the expression of mitoSOX was reduced in the spleens of GRIM19-Tg mice, compared with WT mice.“
Where are these data? I could not find mitotracker and mitoSOX measurements throughout the manuscript.
Comment 8:
Lines 394-395 „Thus, GRIM19 may be a potential therapeutic agent for enhancing mitochondrial activity while downregulating mitochondrial damage“
Before concluding that Grm19 enhances mtochondrial activity, the authros should condcut some sort of mitochondrial activity measurements, e.g. a Seahorse mitochondrial stress test
Comment 9:
Line 364-365: „This study showed that GRIM19 regulates C2C12 differentiation and ameliorates obesity through the regulation of the reciprocal balance between WAT and BAT“.
This is much more complex and several additional measurements would be needed in order to be able to conclude that Grim19 regulates obesity via BAT/WAT differentiation: i) food intake of Grim19-tg and control mice, ii) indirect calorimetry and iii) respiratory quotient. If these data cannot be provided, BAT/WAT differentiation effects seen in vitro are associated with Grim19 efetcs in vivo on body weight - the sentence in Line 364-365 must then be rephrased accordingly
Minor comments:
Line 284-285: „The HFD-fed GRIM19-Tg (GRIM19-Tg 285 HFD) mice exhibited a phenotype similar to that of GRIM19 HFD mice.“ – I do not understand the meaning of this sentence – pls explain
Lines 205-208: Give components and concentrations of the 3T3-L1 cell differentiation medium and 3T3-L1 adipocyte maintenance medium.
Provide the sequence and/or ID of the siRNA targeting mouse Grim19
Provide the catalog and/or clone number for thea nti-Grim1 antibody used for western blotting
Immunoblotting: Provide the ingredients and their concentrations of the lysis buffer
Lines 92-96: The information on the plasmid seems tob e redundant – for generating Grim19-tg mice it seems the same plasmid was used as described in lines 214-215? This can be combined/simplified.
Figure 6A: I suppose the x-axis labeling should show „HFD“ not „NFD“ right?
Reviewer 2 Report
In this study Jhun et al. aim to analyze the role of GRIM19 in white fat browning. To this end they studied WAT and BAT-related genes in the cell lines 3T3-L2 and C2C12 in which GRIM19 was knocked down or overexpressed, respectively. Furthermore, they found that overexpression of GRIM19 partially reduces high-fat diet induced obesity phenotype in mice. In these mice hepatic triglycerides, fat droplets, adipocytes in the epididymal fat and intracapsular lipid vesicles were observed. In addition, in the spleen of GRIM19 overexpressing mice the TH17/Treg balance was shifted towards accumulation of Treg after HFD compared to HFD-treated wildtype control mice.
The topic is of potential interest. However, this study has several weaknesses. The rationale of the study is difficult to follow, since the manuscript lacks important informations about the methods, scientific background, a proper description of the results, and conclusions. In addition, in Figure 5 and 6 the legends do not match the graphs. Furthermore, the study misses unbiased analyses of gene expression, and the rational for the selection of the analyzed genes is not getting clear, and should be explained. Finally, the mechanism, how GRIM19 affects the TH17/Treg balance is weak. At least it should be shown that STAT3/STAT5 activation is altered in these T cells subset.
Major points:
- In general, the results lack explanation, which makes it difficult to follow them and to interpret the data:
- Figure 1+2: What is the difference of the 2C12 cell line compared to 3T3-L1? Why have different readouts been used in Fig.1 and Fig.2? Why have certain genes been analyzed in both cell lines and others not. The rational for this should be explained and the Figures synchronized in terms of cell lines and readouts.
- Figure 2: The authors switch between knockdown and overexpression and use different analyses for knockdown and overexpression. Again the rational should be explained and methods synchronized.
- Figure 3: It is not immediately apparent to the reader that the effect of GRIM19 on obesity is shown using two different strategies of GRIM19 overexpression. This needs to be highlighted in the text and figure legend.
- Figure 1: Oil Red-O is missing
- Figure 2:. Quantification of Oil Red-O staining is missing as well as Oil Red-O staining for knockdown condition
- Figure 1,2,3: The analyzes need to be synchronized to better compare both strategies
- Figure 5: The Figure shows staining of TH17/Treg in the spleen. Staining of these cells in the fatty liver should be added as well as co-staining of pSTAT3/STAT5 in TH17 and Treg
- Figure 6: The experimental set up is not clear. HFD as stated in text and legend or NFD as stated in the figure (A). Please clarify
- A characterization of GRIM19-overexpressing mice under NFD is missing. It would be important to study, if overexpression per se alters the TH17/Treg balance?
Minor points:
- Figure 1+2: information of the house keeping gene that was used is missing
- Figure 3: The statistics for Figure 3A (rate of body weight gain) are missing and the legend lacks the information how it was calculated
- Figure 4: an explanation of why the shown tissues were chosen is missing and should be added in the result section.
- Figure 5: the Legend does not fit to the figure, Westernblot data in p-STAT3 are not convincing
- Figure 5B: amplification is missing
- Figure Legends are incomprehensive due to incomplete or wrong information (e.g. see Fig.5)
- The quality of the histology slides is not sufficient and should be improved
- Starting from Figure 5 the rationale why the authors aimed to investigate TH17/Treg balance and thermogenic effect of GRIM19 is not unclear, and should be explained.
Round 2
Reviewer 1 Report
The authors addressed most of my concerns by proiving new data, as well as textual changes and clarifications. The manuscript has improved substantially.
Comments:
Line 393-394: „Whereas some metabolic disorders and obesity are commonly linked to type 2 diabetes, BAT contributes directly to disease onset and persistence.“ – This sentence is misleading as one might assume that BAT promotes (type 2 diabets?) disease onset. However, the opposite is true i.e. BAT suppresses type 2 diabetes features. The sentence should be corrected
figure 4 is now duplicated in the manuscript shown twice, on page 9 and page 12. Instead, figure 1 is now missing but the figure legend is there. The authors should check carefully figures and figure legends.
Reviewer 2 Report
The authors have successfully addressed all my comments and concern. There is one typo in the modified part, p16 line 370 "GRIAM19" instead of GRIM19.
